# Misinformation about the COVID-19 Vaccine in Online Catholic Media

**DOI:** 10.3390/vaccines11061054

**Published:** 2023-06-01

**Authors:** Verónica Israel-Turim, Valentina Laferrara, Ana Regina Rego, Josep Lluís Micó-Sanz

**Affiliations:** 1Blanquerna School of Communications and International Relations, Ramon Llull University, 08022 Barcelona, Spain; joseplluisms@blanquerna.url.edu; 2Rede Nacional de Combate à Desinformação RNCd, Rio de Janeiro 21941-853, Brazil; anareginaregoleal@ufpi.edu.br

**Keywords:** misinformation, Catholic media, COVID-19 vaccine, fake news, infodemic, health communication, vaccination, news media, digital media

## Abstract

During the COVID-19 pandemic, online media were the most widely used sources of scientific information. Often, they are also the only ones on science-related topics. Research has shown that much of the information available on the Internet about the health crisis lacked scientific rigor, and that misinformation about health issues can pose a threat to public health. In turn, millions of Catholics were found to be demonstrating against vaccination against COVID-19 based on “false” and misleading religious arguments. This research analyses publications about the vaccine in Catholic online media with the aim of understanding the presence of information (and misinformation) in this community. An algorithm designed for each media outlet collected COVID-19 vaccine-related publications from 109 Catholic media outlets in five languages. In total, 970 publications were analysed for journalistic genres, types of headlines and sources of information. The results show that most publications are informative and most of their headlines are neutral. However, opinion articles have mostly negative headlines. Furthermore, a higher percentage of the opinion authors come from the religious sphere and most of the sources cited are religious. Finally, 35% of the publications relate the vaccine to the framing issue of abortion.

## 1. Introduction

The COVID-19 pandemic grabbed the attention of all digital news media, as it matches almost all newsworthy criteria [1,2]. The unpredictability and serious consequences of the spread of the virus led to a considerable increase in the consumption of news [3], which became an essential tool for knowing how to act in the face of the emergency [4]. Media are the most widely used source of scientific information and, on many occasions, the only source of information on science-related topics for the general population [5].

During the COVID-19 pandemic, information consumption on the Internet increased dramatically. In Europe, for example, consumption peaks were recorded on the dates when the first governmental measures were taken to curb the spread of the virus [6]. Despite this, research conducted after the pandemic showed that much of the information circulating on the web lacked scientific rigor [7]. Some revealed that information about COVID-19 was often fuelled by infodemic monikers such as “coronavirus lab” or “5G coronavirus” [8], promoting misinterpretations, misleading information and false news [9].

Health professionals have always faced misinformation, as it has existed since awareness of the media’s influence on public opinion. However, both the digital era and the COVID-19 pandemic have evidenced a substantial growth of fake news or manipulated information. Both factors have made these contents gain even more strength, potentially posing a risk to public health [10].

Such was the extent of misinformation generated around the disease, that it was described by the World Health Organization (WHO) as an “infodemic” and, given that misinformation on health issues can represent a threat to public health [11], the agency asked researchers to help define and understand the scope of high-quality and low-quality information circulating on the Internet [12].

The media has a relevant role in emergency situations, as citizens trust them to provide truthful and evidence-based information [13]. Furthermore, people trust that they do so without causing panic in society [14] and have a commitment to correcting misinformation by filtering content and making available information of proven quality [15].

Likewise, the media is a powerful tool for shaping the public agenda, including during the COVID-19 pandemic. Multiple recent investigations have shown its impact on the public perception of the virus. Some of them reveal that the media played an essential role in slowing the spread of the disease [16,17] and others that they encouraged racial attacks by defining COVID-19 as a “Chinese virus” [18,19]. This demonstrates the need to study media behaviour in situations that require the consumption of contrasted information by the public.

The citizenry tends to seek information that aligns with and sustains their beliefs and ways of understanding reality, seeking evidence that confirms their opinions by consuming information that is consistent with their views, even if it is biased. Previous studies argue that this in turn is intensified by dogmatisms and the difficulty of dealing with complexity, as well as by the tendencies of people with very conservative positions, who present greater resistance to revising their beliefs [20,21]. In the case of Catholic communities, where a wide range of forms can be found, we can detect areas that tend to dogma and conservatism, which is why they represent one of the many areas where disinformation can proliferate.

On their part, digital platforms and social media, through the use of algorithmic recommendations, show each person the information that may be of most interest to them [22,23,24]. Thus, they intensify the tendency to search for information that reinforces everyone’s belief system, thereby enhancing the so-called bubble effect [25]. Previous research has shown that on Twitter, for example, fake news spread faster and further than real news [26,27]. Fake news pushes some psychological buttons. A prominent feature of fake news headlines is emotional provocation, oriented to cause shock, fear, anger, and moral outrage [21]. For this reason, we conducted in this study an analysis of the headlines of the publications made about the COVID-19 vaccine in Catholic media.

Some research, such as that developed by the Massachusetts Institute of Technology [28], points to the virilising potential of narratives containing disinformation in view of their strategically thought-out composition and shaped with content that summons affections such as fear and hate [29], whose political potential ends up forging an uncritical receptivity on the part of the individuals who receive the disinformation.

Disinformation favours the monetisation and profitability of digital platforms, at the same time as it benefits their producers with audience and visibility beyond profitability, and thus, it ends up being a strategy of the business model of each platform which, as Zuboff [30] tells us, work using models of action economy with strategies to attract the attention of users in order to extract behavioural capital; currently, in Zuboff’s opinion, this is the main capital of technological capitalism. However, it is worth underlining that platforms deny such a strategy and see disinformation as an unintended consequence of their business models.

The fact is that the power of misinformation to go viral on digital platforms is exploited to the fullest by antivaccine movements that use fake science as a template and guide for the narratives that reach their audiences. The US Center for Countering Digital Hate conducted extensive research and identified the earnings of the top twelve antivaccine figures in the US and concluded that the market for antivaccine communication and disinformation is highly lucrative [31].

There is a high volume of misinformation regarding COVID-19 vaccines, which has been fuelled by conspiracy ideas such as the nonexistence of the virus, the idea that vaccines contain microchips to control the population, or the implementation of 5G [10], and rumours about their safety for human health [32]. Misleading news about vaccines circulates in digital media and social networks which may differ according to the religious community [32]. In Catholic communities, it has been found (something to be studied in this article) that the vaccine is linked to the use of cells from aborted foetuses. Likewise, in Muslim communities, false claims have been observed regarding the use of traces of pork in vaccine development [33], and in Hindu communities, similar claims have been observed but with the modification that the meat utilised was beef [34]. Previous research indicates that misinformation and fake news present one of the main reasons why people are reluctant to receive vaccines [11,35].

## 2. Materials and Methods

To carry out this research, all publications mentioning the COVID-19 vaccine were collected from 109 Catholic websites in five different languages: 30 in Spanish, 15 in French, 26 in English, 18 in Italian and 20 in Portuguese (see Table A1 in Appendix A). All publications referring to the COVID-19 vaccine before 1 April 2021, the date on which Google News Initiatives awarded the Vaccine and Prejudices in the Catholic Community project to Aleteia.org and its entire consortium, were collected.

The collection of posts was conducted using an algorithm designed based on each website studied to collect all posts that, up to 1 April 2021, included certain keywords related to the COVID-19 vaccine. Since the algorithm retrieved some duplicate publications (because of edits in the original publications), a manual monitoring of each of the automatically collected publications was performed. Ambiguous content or content that did not meet the objective of this research was discarded. A third check was performed by the analysts, who verified that each publication in the sample addressed the COVID-19 vaccine topic, while also verifying that it was published within the established study period. This resulted in a total of 970 publications, which were then analysed quantitatively and qualitatively. The quantitative method was used to identify the basic data of the publications and the qualitative method was used to study the journalistic genres, headlines, and the publication in general. The sample was recorded in spreadsheets where, in addition to quantifying the total number of publications collected, categories of analysis were proposed. The categorisation scheme used in this study was based on categorisations used in previous studies [36,37]. The categories for the analysis of the publications (as a whole) were developed exclusively for this research, considering the unique characteristics of the field under investigation. The tool used for categorisation was Google Forms. After identifying the publications referring to the COVID-19 vaccine and categorising them according to the basic identification data (see Table 1), the analysis focused on the journalistic genres (see Table 2), the headlines (see Table 3), and the publication as a whole (see Table 4). The analysis was applied to all publications equally. Difficulties encountered during the process of classifying some publications into categories were resolved using the collaborative cross-checking strategy [38].

After having studied the publications with the forms, we proceeded to work with the data. Through quantitative analysis and the use of data visualisation techniques, we looked for trends and patterns [39,40,41] that could reveal the types of publications most used by these media, the genres and formats, the most recurrent themes and the most used sources of information. This analysis is reflected in the following Results section.

## 3. Results

### 3.1. Origin of the Publications

Of the 968 posts retrieved by the algorithm, more than half came from English-language websites (62.2%). Almost a third were retrieved from Spanish websites (27.6%) and significantly fewer from Italian (5.6%), French (2.8%) and Portuguese (1.7%) websites. Of the 109 websites consulted, publications were retrieved from 58 websites. The sites from which the most publications were retrieved are *Crux* (135 articles) and *ACN web* (102). Almost half of the publications were retrieved from *Catholic News Agency* (61), *American Magazine* (53), *Catholic Philly* (51), *National Catholic Register* (51) and *Catholic Newspaper* (51). Fewer than 50 publications were retrieved from the remaining websites (Figure 1).

The publications were disseminated by sites from 17 countries. Four of the studied sites were classified as “Global” because they are based in more than one territory and are characterised by their international reach. This is the case of *Aciprensa*, *Gaudium Press*, *Catholic News Service* and *Cisa News Africa*. About half of the publications retrieved, and thus analysed, (48.7%) come from US websites. The remaining half is distributed among the rest of the territories detected, including the “Global” category (Figure 2).

### 3.2. Support for Websites

Most of the publications correspond to media whose original support is the Internet (58%) (Figure 3). This can be observed in the publications of all languages, except for French-language publications, where 74% of the publications correspond to media whose original support is the press. In the remaining languages, Internet media publications correspond to 54% of English publications, 57% of Spanish publications, 85% of Italian publications and 100% of Portuguese publications. The only language in which media whose original support is television was identified is English (7%) (Figure 4).

### 3.3. Publication Date

Throughout the study period, peaks of information production on the COVID-19 vaccine can be observed. When analysing the total number of publications, two dates show the highest number of articles published on this topic, with English and Spanish websites producing the highest volumes of publications on these dates (Figure 5).

21 December 2020

This news spike responded to a statement issued by the Congregation for the Doctrine of Faith, approved by Pope Francis, in which he announced: “All vaccines recognised as clinically safe and effective can be used in conscience with the certainty that the use of such vaccines does not constitute a formal cooperation with the abortion from which the cells used in the production of the vaccines are derived”. The text stated that the reason is that the abortion from which the cell lines were extracted “is, on the part of those who make use of the resulting vaccines, remote”. However, the Congregation clarified that this thumbs-up does not equate to approval of abortion or the use of cell lines derived from aborted foetuses in scientific testing.

23 February 2021

Among Spanish-language publications, several unrelated issues are identified, so the increase in news output is believed to be rather coincidental. In English publications, however, two recurring themes can be observed. One is the call by Europe’s Catholic leaders (Caritas Europa and the Commission of Bishops’ Conferences of the European Union) for the EU to be guided by solidarity, fraternity, and social justice in distributing and administering the COVID-19 vaccine. “Commitment and solidarity must be the decisive criterion at this historic moment,” they said in a joint statement. The other, meanwhile, was the prayer offered by Cardinal Wilton D. Gregory of Washington for those who had died of COVID-19 on CNN’s programme marking 500,000 deaths from the disease.

Three informative peaks can be observed in the Spanish publications:14 December 2020

Publications on this date are linked to the start of vaccination against COVID-19 in the United States, with a nurse in New York City being the first person to receive the vaccine. This event prompted a joint statement from the chairs of the Doctrine and Pro-Life Committees of the United States Conference of Catholic Bishops, in which they argued that vaccination is “a moral responsibility”. Receiving one of the COVID-19 vaccines should be understood as an act of charity towards other members of our community. “In this way, being safely vaccinated against COVID-19 should be considered an act of love of neighbour and part of our moral responsibility for the common good,” wrote prelates Kevin C. Rhoades and Joseph F. Naumann, while not obviating the “moral concerns” of vaccines. The bishops of the state of Colorado also spoke out, stating in a letter that the use of vaccines is morally acceptable under certain circumstances. In this sense, they recalled that “a good end cannot justify bad means” and that vaccines “must be developed according to ethical criteria”. The Canadian government warned that those allergic to any ingredient should not use the Pfizer vaccine. Another event that was identified in the publications on this date was the demonstration of some twenty COVID-19 denialists around Seville Cathedral. “The mask makes us sick”, “COVID-19 vaccine = genetic engineering” and “The vaccine kills you” were some of the messages of the demonstrators.

21 December 2020

Postings on this date relate to the Vatican’s declaration that it is morally acceptable to vaccinate against COVID-19. 55% of the publications on this date report the content of the Vatican’s statement with headlines that refer to the vaccine in a positive or neutral way, such as “Holy See calls COVID-19 vaccines “morally acceptable””, or “Moral assessment of the use of COVID-19 vaccines”. The remaining 45% tend towards the negative tone and criticism of this note issued by the Vatican, either with headlines such as “Christians should never take vaccine contaminated by abortions, says bishop” or “Vatican applies vaccine contaminated by abortions”. You can also find headlines on this date where no reference was made to the fact of the note issued by the Vatican, but in which the vaccine was associated with negative side effects and even death from the vaccine.

11 March 2021

On this date, although a peak of publications can be observed, they are not related to each other. Nor has it been possible to find an informative event or fact that justifies the increase in publications on this date, so it could be considered a chance occurrence, with no identified causality.

In the English-language publications, a greater number of information peaks about the COVID-19 vaccine are identified, which coincide with the following dates:14 January 2021

Growth in publication volume is recorded due to Catholic websites reporting that a Vatican spokesperson confirmed that Popes Francis and Benedict XVI received the first dose of the COVID-19 vaccine on the previous Wednesday and Thursday, respectively.

18 February 2021

Most publications on this date report a decree issued by the Cardinal Governor of Vatican City State warning that employees who refuse to receive the COVID-19 vaccine could face sanctions or dismissal.

4 March 2021

On this day, the chairman of the US bishops’ Committee on Doctrine, Bishop Kevin C. Rhoades, released a video in which he said the Johnson & Johnson vaccine “can be used with a good moral conscience”. The bishop said the Vatican “has made it clear that all COVID-19 vaccines recognised as clinically safe and effective can be used in good conscience”. He also said that if there is a choice of vaccines available, “we recommend that you choose one with the least connection to abortion-derived cell lines”.

10 March 2021

On this date, a group of leading Catholic academics announced that they believed it was morally acceptable for anyone to receive any of the vaccines available in the United States. “Catholics, and indeed all people of good will who embrace a culture of life for the whole human family, born and unborn, can use these vaccines without fear of moral culpability” for abortion, they said. The variation in the news output of the Portuguese, French and Italian websites is not significant, as in no cases are their peak days more than three publications. Therefore, no correlations with specific current events were established.

## 4. Journalistic Genre of the Publications

Regarding the journalistic genres, in the news sites analysed, publications of the informative genre had the greater presence, representing 90%, compared to 10% of the opinion genre. Comparing by language, publications in Italian contained the highest percentage of the opinion genre (14%). In the rest of the languages, publications in this genre are between 7% and 13%. In all the languages, publications in the informative genre are the majority, exceeding 85% (Figure 6).

### 4.1. Informative Genre

#### 4.1.1. Informative Subgenres

Among the publications of the informative genre, 83% are news, that is, journalistic texts that consist of the narration of a fact or current event. This is followed by the report subgenre with 10%, these being journalistic texts that include news elements and declarations of various personalities, and which are mainly descriptive. To a much lesser extent, 4% were identified as chronicles, which are composed of interpretative accounts told from the scene of the events, including evaluative elements. Finally, there were 3% identified as interviews, i.e., dialogues with people who can provide data on a particular event.

It can be observed that news reports and chronicles were only identified among publications in Spanish and English. In addition, only news items were found in Portuguese-language publications. This subgenre was the majority in all the studied languages, representing more than 80% of the publications in all the languages (Figure 7).

#### 4.1.2. Type of Informative Headlines

Most of the headlines of the publications identified as informative corresponded to the referential type (93%), that is, of an objective order, insofar as they are true or not false (see Figure 5). Examples of this type of headline are:In Spanish: “Biólogo molecular católico: Vacunas contra el COVID-19 son seguras y eficaces” (*Aciprensa*, 11 January 2021);In English: “‘We need the Lord to cast out demon of division in our nation’, cardinal says” (*Catholic News Service*, 14 January 2021);In Portuguese: “Anvisa aprova registro definitivo da vacina de Oxford no Brasil” (*Canção Nova*, 12 March 2021);In French: “COVID-19: «La dimension géopolitique est centrale dans la course au vaccin»” (*La Croix*, 8 December 2020);In Italian: “Coronavirus COVID-19: vescovi statunitensi, ‘dubbi su ammissibilità morale uso vaccini sviluppati, testati o prodotti con linee cellulari derivate da aborto’” (*Agensir*, 3 March 2021).

The remaining 7% of the informative headlines were valuative, that is, they were of an interpretative nature, insofar as they reflect an interpretation of the journalist or author of the publication about current affairs, in accordance with his or her values, interests or ideology. Examples of this type of headline are:In Spanish: “Los bebés por nacer utilizados para el desarrollo de las vacunas estaban vivos en el momento de la extracción de tejido” *(ACN web*, 13 January 2021).In English: “Aborted baby vaccine backfires” (*Church Militant*, 10 August 2020).In French: “COVID-19: un vaccin, un espoir et des questions” (*La Croix*, 13 January 2021).In Italian: “Vaccino anti COVID: perché si continua a fare allarmismo?” (*La Luce di Maria*, 29 January 2021).

When compared by language, publications in French used the greatest number of valuative headlines (24%). In publications in Spanish (7%), English (6%), and Italian (4%), a lower use of this type of headline was detected, while no publications in Portuguese were identified with this type of headline (Figure 8).

### 4.2. Opinion Genre

#### 4.2.1. Opinion Subgenres

The largest number of opinion publications on the COVID-19 vaccine were in the column format (81%), which means that they were signed and offered a point of view or disquisition on the subject in question. This type of publication predominated in all the languages studied, since in all of them they represent almost or more than 80% of the opinion publications and in both Portuguese and French, 100% (Figure 9).

The second largest format in the total number of opinion publications was editorial (17%), that is, unsigned publications that express the opinions of the media itself, which do not contain personal positions, but rather those of the collective intellectual who is behind the publication of the website. This format, however, was only present in opinion publications in Spanish and English, where it represented 20% and 19%, respectively (Figure 9).

Opinion publications in the reader’s letter format appeared in a much lower percentage (2%); these publications are written by the readers of the media and are intended to express their opinion on the topic in question. This format was identified in 12.5% of the publications in Italian and 1.7% of the publications in English. It was not detected in the publications in Spanish, Portuguese and French (Figure 9).

#### 4.2.2. Opinion Headline Types

More than half of the headlines of opinion publications were of a valuative nature (61%), so it can be said that they include an interpretation, evaluation, or judgment of the journalist on certain facts of reference, while attributing a contextual meaning to them. The remaining percentage (39%) was thematic, which means that they only indicate the issue addressed in the publication.

In all languages, half or more than half of the opinion publications were of an evaluative type, with Spanish publications having the highest percentage of this type of headline (64%). In Portuguese and French publications, an equal distribution is observed: 50% of opinion publications had thematic headlines and 50% were evaluative (Figure 10). However, the results for Portuguese and French cannot be considered representative given that only two opinion publications were detected in each language.

Sixty one percent of the evaluative headlines corresponding to opinion publications were negative. A breakdown of the evaluative headlines by language shows that opinion headlines in Spanish (87%), French (100%) and Italian (100%) contained the highest percentages of negative evaluative headlines. Conversely, in English (57%) and Portuguese (100%), there were mostly positive evaluative headlines (Figure 11).

#### 4.2.3. Opinion Authors’ Profile

Most of the authors who wrote opinion articles have a journalistic profile (34%) linked to the media, or a religious profile (18%) linked to the Catholic community. A smaller percentage of opinion publications were signed by an author with expertise in different disciplines (13%). A significant percentage of authors have a mixed profile (28%), that is, they are linked to more than one of the specified fields. To a much lesser extent, there are profiles related to other fields (4%), and other (3%).

This trend is observed in both Spanish and English publications. However, this does not occur in Portuguese, French and Italian publications (Figure 12). In Portuguese, the authors identified are mainly linked to the fields of experts in different disciplines (50%) and religion (50%). In French, all the authors have a mixed profile. Finally, in Italian, the largest number of authors are linked to the media (63%), followed by the fields of experts in health sciences (25%) and religion (12%). Once more, it is important to note that the results for Portuguese and French publications cannot be considered representative given that only two opinion publications were identified in each language.

### 4.3. Subject of the Publication

The most recurrent theme among the total number of publications analysed was the access to COVID-19 vaccines, i.e., their distribution among the population (Figure 13). The publications identified under this thematic category raised issues mainly linked to the ability to provide equal opportunities in access to the vaccine. Among the main information related to this topic, we can find the repeated petitions from Pope Francis for the inclusion of the entire population in the vaccination campaigns against COVID-19, paying special attention to providing access to vaccines to the most vulnerable populations. Information on the vaccination campaigns carried out by the Vatican for these populations also stands out. An example of this is the publication made on 26 March 2021, by *AICA*, entitled “The initiative is carried out by the Apostolic Limosneria and responds to the repeated appeals of Pope Francis so that no one is excluded from the vaccination campaign against COVID-19”.

The second most recurrent theme was information and misinformation on COVID-19 vaccines, i.e., publications that aimed to provide useful data on vaccines. The third most frequent topic was related to vaccine composition. The rest of the categorised topics occupied a considerably smaller space.

Although most of the publications analysed did not link COVID-19 vaccines to abortion (69%), a third of them did (31%). This link proposed that vaccines are composed of cells from aborted foetuses and/or that they can cause spontaneous abortions.

A difference is observed between the percentage of publications that, in each language, linked vaccines against COVID-19 with the topic of abortion. The language with the highest percentage of publications linking COVID-19 vaccines to abortion was English (38%). This was followed by publications in Spanish (26%) and, to a much lesser extent, in Italian (9%), French (7%) and Portuguese (6%) (Figure 14). It could be relevant to state in this regard, that most of the publications in English corresponded to U.S. media (81%) and most of the publications in Spanish (41%) to Mexican media; further exploring such correlations could represent an interesting line of future research.

### 4.4. Sources of Information

The most cited sources of information among the analysed publications came from the health field. They corresponded to health personnel and experts in health sciences. The second type of sources most frequently used came from the religious sphere, i.e., from the Catholic community.

Twenty percent of the publications that addressed the subject of the composition of the vaccine contained only religious sources, and 76% contained at least one of them, even though they also contained other types of sources. Only 4% of the publications on this topic used only health personnel and health science experts as sources, and 58% included them as one of the types of sources (see Table 5).

Thirteen percent of the publications whose subject is vaccine information/disinformation contained only religious sources, while 2% mentioned only sources of information from health personnel and experts. However, 61% of the publications dealing with this topic contained at least one source of this type, among others. This same percentage corresponds to the use of religious sources among other types of sources (see Table 6).

## 5. Discussion

The results of the conducted research reveal that disinformation also reached the Catholic community through its own communication vehicles, and considering the five languages analysed, highlight the complexity of the phenomenon in the midst of the pandemic process and the potential health risks for this population. Disinformation in this context goes beyond a market that runs through digital platforms and social networks and is established within Catholic journalistic platforms, where the ethics and aesthetics of journalism are in line with Catholicism, emphasizing aspects of personal and collective belief in an environment marked by the undeniable word of God. In this research, comparisons were made by language, considering those mostly spoken by catholic communities, but not by country or region. However, this may also bring interesting results to compare with the present study, and thus is a suggested line for future research.

Moreover, the results of this study are consistent with previous research that has shown that the COVID-19 pandemic, like any pandemic [1], has become a global phenomenon that has captured the attention of digital media around the world [2,4]. The algorithm created for this research scrapped 109 Catholic media outlets in five different languages: Spanish, English, French, Portuguese and Italian. In total, it collected 968 publications from 58 websites. More than half of the publications retrieved by the algorithm and, therefore, analysed in this research, came from websites in English (62%) and a little less than a third (28%) from websites in Spanish. Fewer publications were from Italian (6%), French (3%) and Portuguese (2%) websites. The sites from which the largest number of publications were retrieved are *Crux* (135) and *ACN web* (102). In addition, noteworthy numbers of the publications were analysed from *Catholic News Agency* (61), *American Magazine* (53), *Catholic Philly* (51), *National Catholic Register* (51) and *Diario Católico* (51). Less than 50 publications were recovered and analysed from the rest of the websites covered in this research. Moreover, almost half of the publications analysed (49%) come from U.S. websites (Figure 2). These results suggest a higher attention given by the mentioned Catholic media outlets and languages to the vaccine topic. However, this may also be because the algorithm was more successful in retrieving vaccine-related content from some media than from others and, as a result, collected a greater number of publications from those media. Therefore, the results should be interpreted with caution.

The results show peaks of informative production on COVID-19 vaccines (Figure 5). These peaks are registered in the Spanish and English publications, as the variation in the information production of the Portuguese, French and Italian websites was not significant since it did not exceed three daily publications on the days of highest production. All the peaks identified coincide with events linked to COVID-19 vaccines that can be considered relevant to the Catholic community.

Regarding the journalistic genre, the most frequently used was the informative genre, accounting for 90% of the publications, compared with 10% opinion publications. The language in which the opinion genre obtained the highest percentage was Italian, with 14% of publications, followed by Portuguese publications, that reached 13% of this type of publications. Likewise, in all languages, informative publications accounted for more than 85% (Figure 6); French was the language with the highest percentage of informational publications (93%), followed by Spanish informative publications with 91%, and English with 90% informative publications.

Among the publications of the informative genre, we find that the majority are news (83%). In second place comes the report (10%) and then, to a much lesser extent, the chronicle (4%) and the interview (3%). Reports and chronicles were only identified among publications in Spanish and English, while publications in Portuguese were only news.

Most of the headlines among informative publications were referential (93%), meaning that they intended to be objective, and they are true or not false. A minor percentage of informative headlines (7%) were of an evaluative type (valuative), which embody an interpretative order insofar as they reflect an interpretation of the journalist or author of the publication in accordance with his or her values, interests, or ideology. French-language publications used the greatest number of valuative headlines (24%) (Figure 8).

Within the opinion publications, 81% were columns, i.e., signed journalistic texts that offered a point of view or disquisition on a particular topic. Columns prevailed as the majority of the opinion publications, with at least an 80% in all the studied languages, and in the cases of Portuguese and French, the opinion genre represented 100% of the publications (Figure 9). The second most used format, with 17%, was the editorial, namely, unsigned publications that express the opinions of the media outlet itself, which do not contain personal positions, but those of the intellectual collective behind the website’s publication. This format, however, was only identified in opinion publications in Spanish and English. To a much lesser extent, opinion publications appeared in the format of a reader’s letter (2%).

Most of the opinion headlines were of a valuative nature; a total of 61% of the headlines of publications of this type included an interpretation, evaluation, or judgment by the journalist with respect to certain facts of reference. Thus, less than 40% of opinion headlines were thematic, indicating only the issue addressed in the publication. When dividing per language, this trend can also be observed among Spanish and English publications (64–36% and 59–41%, respectively). On their hand, Portuguese and French publications presented a 50–50 distribution of thematic and valuative headlines, whereas Italian presented 75% valuative headlines in contraposition to 25% of the thematic type. It is interesting to note how the Italian language presents some differences with regards to the rest of the languages. On the one hand, it had the highest percentage of opinion publications (14%), and within opinion publications, it had the highest percentage of evaluative headlines (75%). This result could suggest a more explicit tendency from the Italian Catholic media to present their opinions when referring to the COVID-19 vaccine, while the media analysed in English or Spanish, for example, showed a greater tendency towards communicating vaccine-related issues in a format that seeks to present (or pretend) neutrality and information, even when false information is given, as in the case of linking the vaccine to abortion. In any case, even counting Italian publications, the majority of publications in all languages tended to be presented in an informative format, representing more than 80% across all languages. Moreover, within the informative publications, the vast majority were of the news type and had referential headlines; therefore, the tendency of all the Catholic media analysed is to present information on the COVID-19 vaccine with facts in a neutral referential news format.

Another result that is worth highlighting is the fact that within the informative publications, French publications presented 24% evaluative headlines. The language that used the most headlines of this type after French was Spanish with 7%, which shows a difference between French and the rest of the languages. Sixty-one percent of the evaluative headlines had a negative connotation. However, this majority is not observed in all languages. The language division shows how this majority is observed in French and Italian evaluative headlines, in 100% of the cases, and in the Spanish evaluative headlines, in 87% of the cases. However, in evaluative headlines in English, 57% of the headlines contained a positive connotation, and in the case of the evaluative headlines of the opinion publications in Portuguese, 100% were positive (Figure 11).

When analysing the profiles of the authors of the opinion publications, it can be observed how the majority have a journalistic profile, with a 34% of authors belonging to the media sector. A mixed profile represents 28% of opinion publications’ authors, 17% of these publications were written by authors with a religious profile, and 13% of the profiles belong to experts in different disciplines. The present study did not analyse whether author profiles had any correlation with the tone, stance or persuasiveness of opinion publications, nor the impact of each author on the recognition of his or her publications, an interesting line of research to explore in the future.

Access to COVID-19 vaccines and the capacity to provide the vaccine was the subject of most of the publications of all types. The second most frequent subject was the misinformation regarding the COVID-19 vaccines, and the third, the composition of the vaccines (Figure 13). Regarding such composition, a link between the COVID-19 vaccine and abortion was detected in more than a third of the 968 analysed publications. In the same way previous research had found misleading news about vaccines circulating through the digital sphere among different religious communities [32], in the present investigation it was found that 31% of the publications suggested that the COVID-19 vaccines either contained cells from aborted foetuses or caused spontaneous abortions. The language in which this linkage occurred to the greatest extent was English, where 38% of the publications presented this linkage. In second place were the publications in Spanish, with 26% of the publications (Figure 14). In line with previous research that demonstrated that there is a high volume of misinformation regarding vaccines for COVID-19 and conspiracies and mistrust about their health safety [10,32], the present research has found misinformation linking the vaccine to the use of cells from aborted foetuses and/or causing miscarriages.

With regards to the sources of the publications, the type of source most frequently cited corresponds to health personnel and experts in health sciences. In the second place, we see sources coming from the religious sphere. Among the publications that communicated about vaccine composition, one-fifth contained only religious sources, and in 76% of these cases, some of the sources were religious even if they included other types of sources. Four percent of the publications on this topic used health personnel and health science experts as their only source, and 58% of these publications used at least one such source (Table 5). Likewise, within the publications that informed about vaccine information/disinformation, 13% had religious people as their only source and 2% had only health care personnel and experts as sources, but 61% of them used at least one source of this type. Equally, in 61% of these publications, we found at least one religious source (Table 6).

In conclusion, the tendency among the Catholic media analysed was to present the information related to the COVID-19 vaccine in an informative, newsworthy format with referential headlines, using health and religious sources. Likewise, the topic that most occupied the analysed publications was access to the vaccine and the ethics or morality of getting vaccinated. This study focused on the news production from Catholic media, which opens a clear future research venue with focus on the consumption of these media and the influence of their formats on readers’ attitudes, as well as the role that different formats play in shaping their perceptions and beliefs. Overall, this research opens a deeper reflection on what happens when journalism and religion meet, the former being a field in which professionals are expected to convey facts based on evidence and research [13,14,15].

## Figures and Tables

**Figure 1 vaccines-11-01054-f001:**
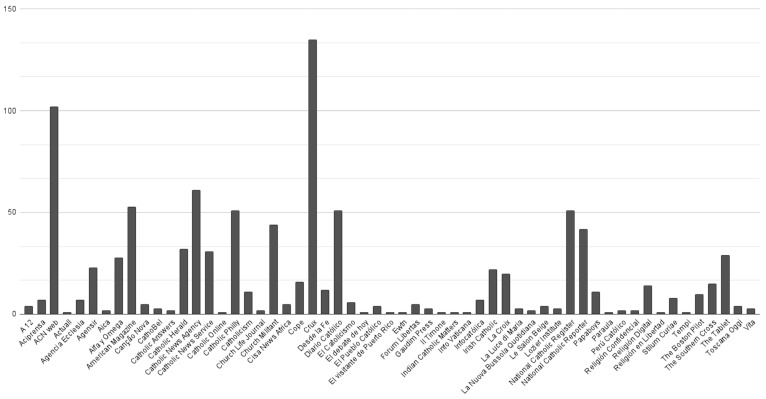
Number of publications per media outlet.

**Figure 2 vaccines-11-01054-f002:**
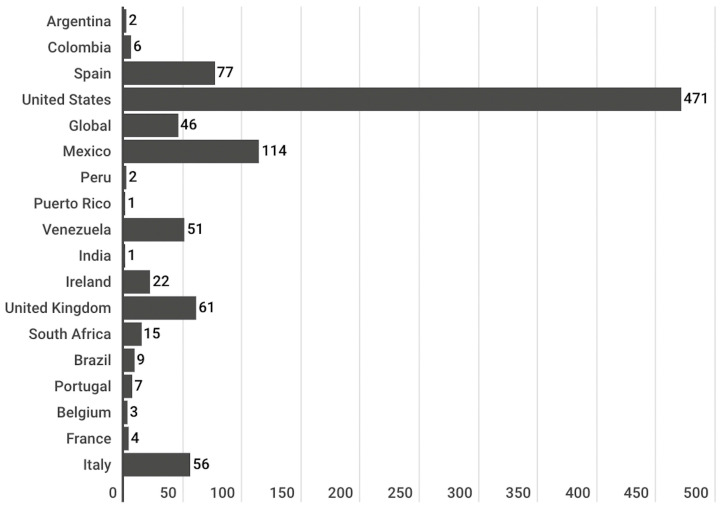
Countries of origin of publications.

**Figure 3 vaccines-11-01054-f003:**
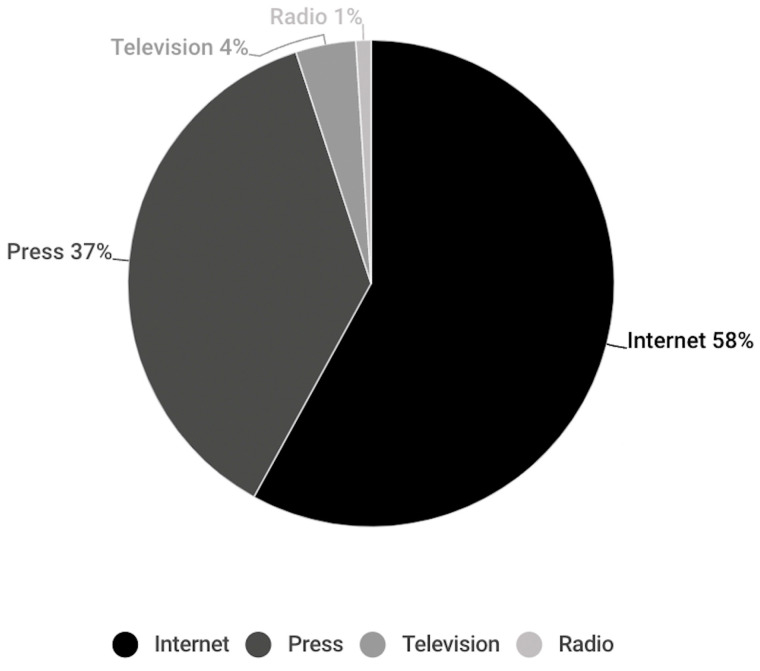
Support for the total number of publications analysed.

**Figure 4 vaccines-11-01054-f004:**
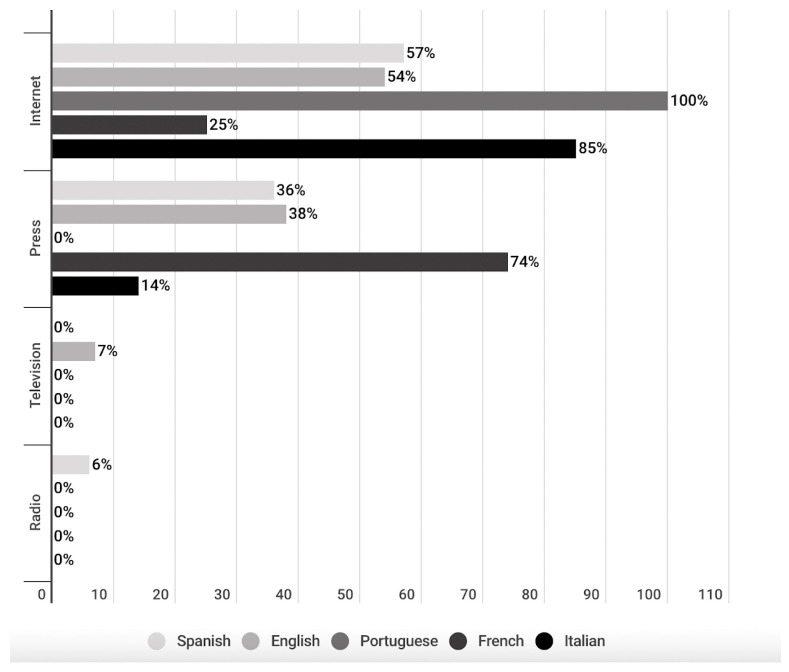
Support of publications by language.

**Figure 5 vaccines-11-01054-f005:**
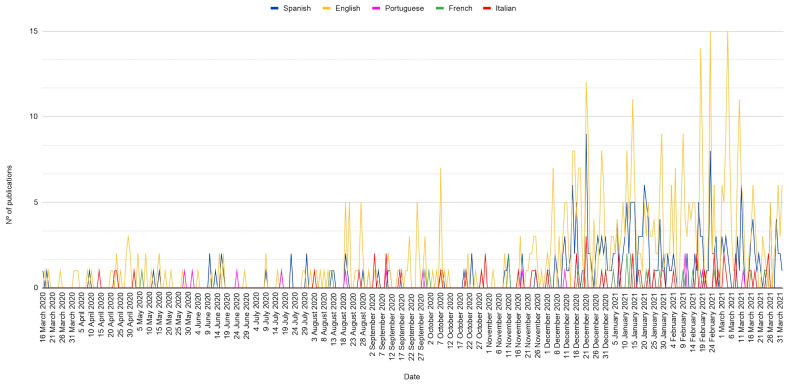
Comparison of the evolution in the information production of the studied websites.

**Figure 6 vaccines-11-01054-f006:**
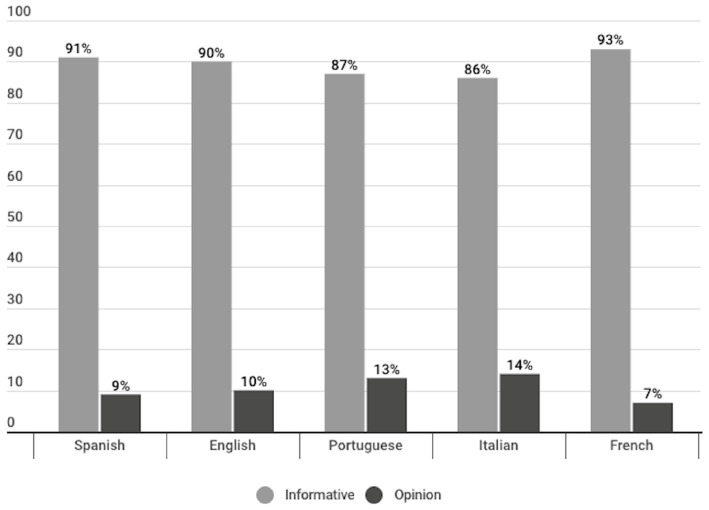
Journalistic genre of the analysed publications, divided per language (%).

**Figure 7 vaccines-11-01054-f007:**
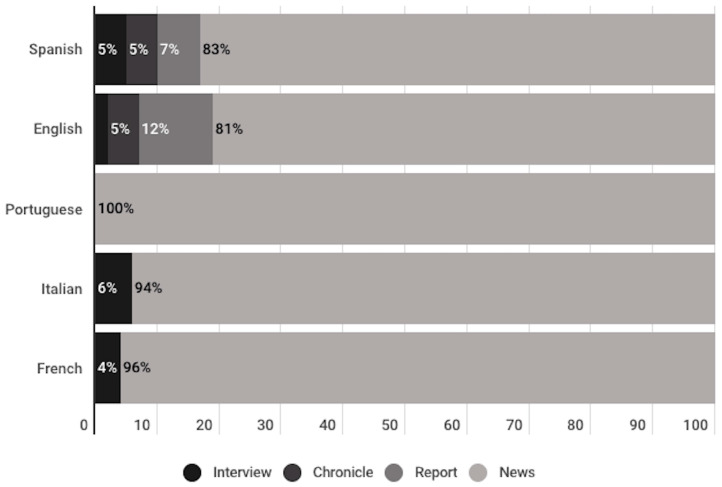
Journalistic subgenre of informational publications, divided per language.

**Figure 8 vaccines-11-01054-f008:**
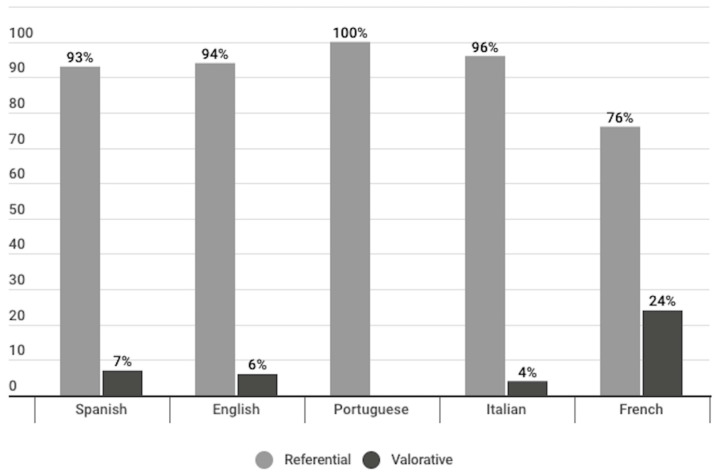
Types of headlines of the informative publications, divided per language.

**Figure 9 vaccines-11-01054-f009:**
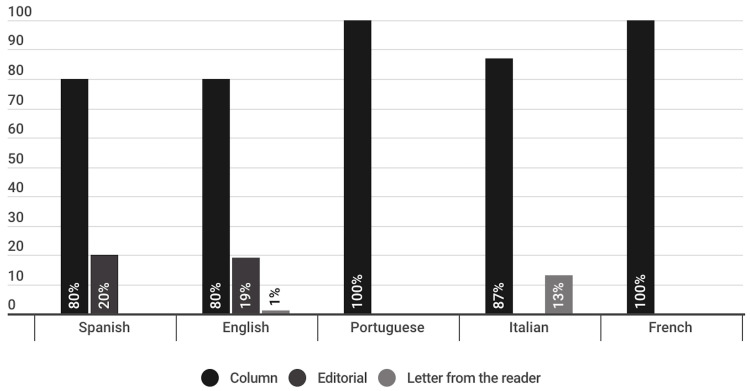
Journalistic subgenres of publications classified as opinion, divided per language (%).

**Figure 10 vaccines-11-01054-f010:**
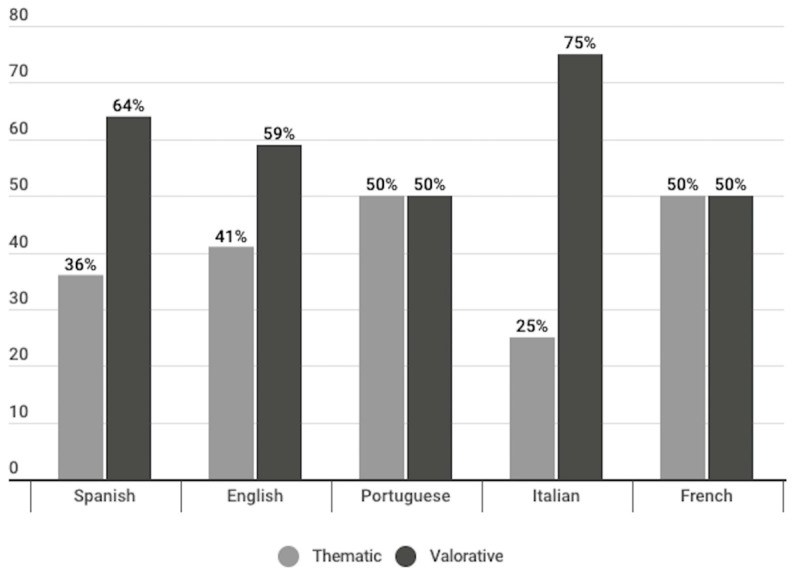
Types of headlines of the publications identified under the opinion genre, divided per language (%).

**Figure 11 vaccines-11-01054-f011:**
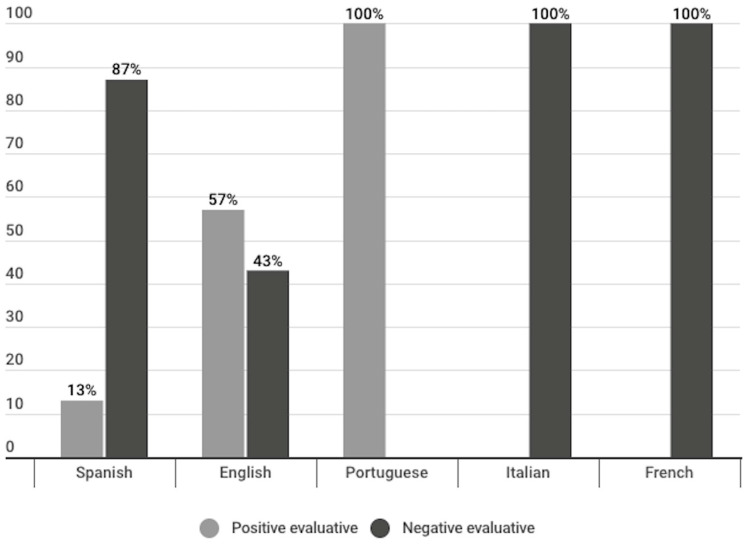
Type of evaluation of the headlines of opinion publications, divided per language (%).

**Figure 12 vaccines-11-01054-f012:**
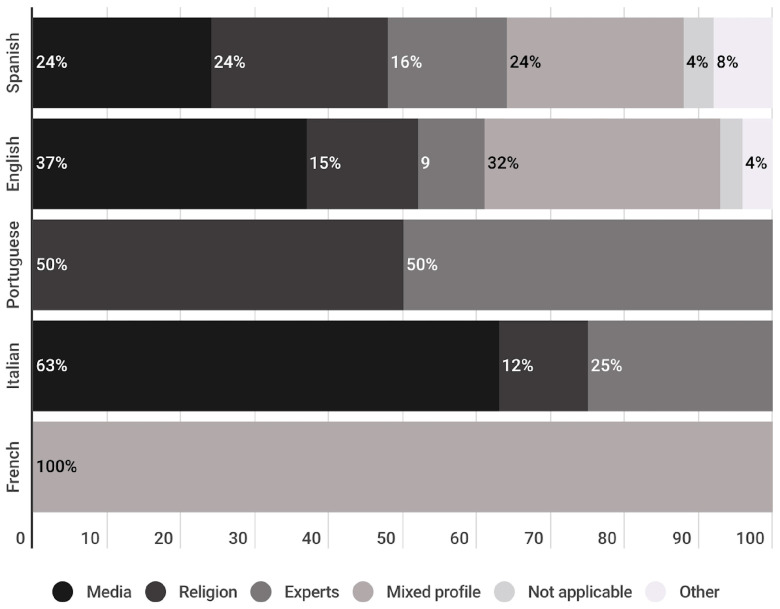
Profiles of the authors of the publications of the opinion genre, divided per language (%).

**Figure 13 vaccines-11-01054-f013:**
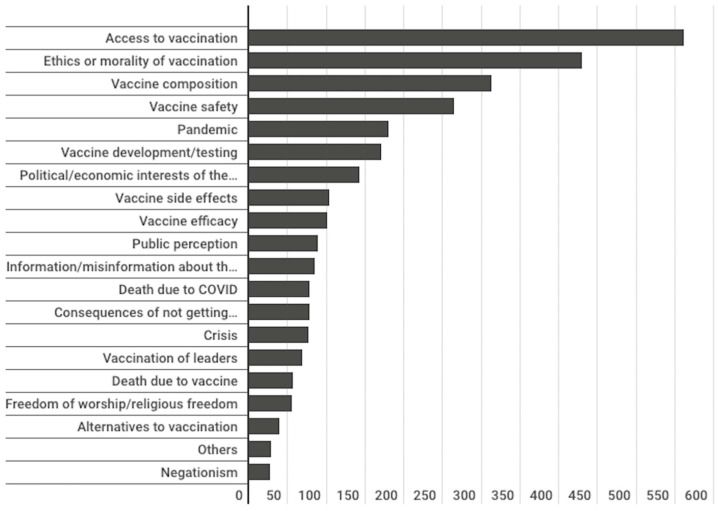
Publications’ subjects.

**Figure 14 vaccines-11-01054-f014:**
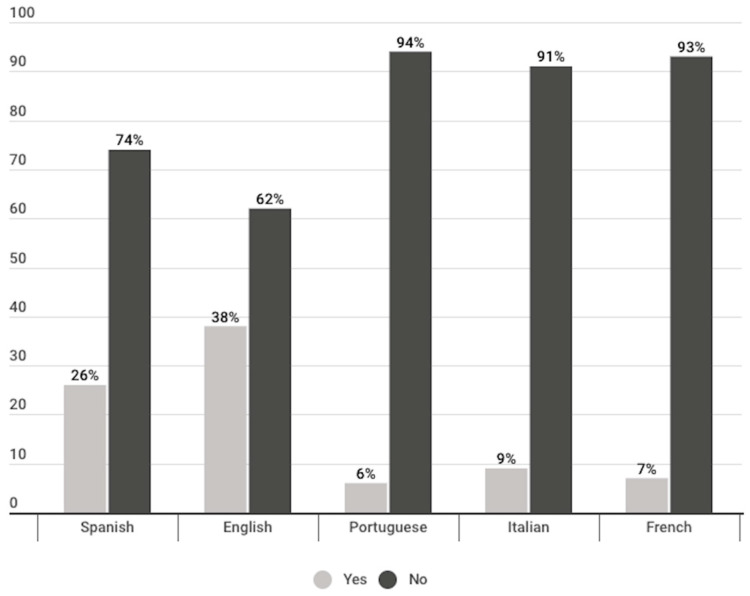
Linkage of COVID-19 vaccine to abortion, divided per language (%).

**Table 1 vaccines-11-01054-t001:** Categories for the analysis of the publications’ basic data.

Categories	Subcategories	Description
Language	Spanish	-
	French	-
	English	-
	Italian	-
	Portuguese	-
Country	-	-
Media	-	-
Publication date	-	-
Headline	-	-
Signature	Author	Signed by an identified author
	Newsroom	Signed by the newsroom staff
	Unidentified	The author is not identified
Original media support	Television	-
	Radio	-
	Press	-
	Internet	-
Journalistic genre	Informative	Facts or current events predominate.
	Opinion	The author’s point of view predominates, regardless of whether it refers to facts or current events.

**Table 2 vaccines-11-01054-t002:** Categories for the analysis of publications according to genre. Source: [36].

Categories	Subcategories	Description
Informative	Chronicle	Journalistic text based on an interpretative account and told from the place where the news events took place. It includes some evaluative elements.
	Interview	Journalistic text based on a dialogue with the aim of obtaining information that someone can provide about a fact or to get to know the personality of a certain person.
	News	Journalistic text that consists of the narration of a fact or current event.
	Report	Journalistic text that includes news elements, declarations of different characters or testimonies and that, fundamentally, has a descriptive character. It is usually based on a recreation of a news event.
Opinion	Column	Text offering an opinion or point of view on a current issue, or a disquisition by the author himself.
	Reader’s letter	A type of opinion letter written by readers to express their opinion on a current news item, or to denounce or support an event.
	Editorial	Unsigned text that expresses the opinions of the media outlet itself. It does not contain personal positions, but those of the collective intellectual behind the publication of the newspaper or magazine.

**Table 3 vaccines-11-01054-t003:** Categories for headline analysis. Source: [37].

Categories	Description	Subcategories
Referential	Objective in that they are either true or not false. They are not globally interpretative or evaluative.	Subject of the headline
		Topic of the headline
		Sub-topic of the headline (in case the subject was the COVID-19 vaccine)
		Way of referring to the subject (positive, negative, neutral)
Valuative	Interpretative in that they reflect a journalist’s interpretation of current affairs, in accordance with his or her values, interests or ideology.	Type of valuation (positive, negative, neutral)
		Subject of the headline
		Subject affected by the valuation
		Topic of the headline
		Sub-topic of the headline (in case the subject was the COVID-19 vaccine)
		Way of referring to the vaccine (positive, negative, neutral)

**Table 4 vaccines-11-01054-t004:** Categories for the analysis of the publication (as a whole). Source: Own elaboration.

Categories	Subcategories
Subject	Vaccine composition
	Side effects
	Ethics or morals of vaccination
	Political/economic interests of the vaccine
	Religion
	Religious freedom
	Negationism
	Public perception
	Other
Link between vaccine and abortion	Yes
	No
Coherence between title and text	Yes
	No
Sources of information	News agencies
	Media
	Other sciences (theologians, sociologists, psychologists, etc.)
	Health personnel and health science experts
	Politics
	Religion
	Society/citizenship
	No sources

**Table 5 vaccines-11-01054-t005:** Percentage of publications on the subject “Composition of vaccines” that contained religious and health sources.

Publications Mentioning Only Religious Sources (%)	Publications Mentioning Religious Sources among Other Types of Sources (%)	Publications Mentioning Only Sources of Health Personnel and Health Experts (%)	Publications Citing Sources from Health Personnel and Health Experts among Other Sources (%)
20%	76%	4%	58%

**Table 6 vaccines-11-01054-t006:** Percentage of publications in the subject area “Vaccine Information/Disinformation” that contained religious and health sources.

Publications Mentioning Only Religious Sources (%)	Publications Mentioning Religious Sources among Other Types of Sources (%)	Publications Mentioning Only Sources of Health Personnel and Health Experts (%)	Publications Citing Sources from Health Personnel and Health Experts among Other Sources (%)
13%	61%	2%	61%

## Data Availability

Not applicable.

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
