# Peer review of "Misinformation about the COVID-19 Vaccine in Online Catholic Media"

_vaccines, 2023, doi:10.3390/vaccines11061054_

Round 1

Reviewer 1 Report

An interesting manuscript that required an unusual amount of investigative search and interpretation to put it together. Not quite the type of info that I am accustomed to reading in the Journal VACCINES but maybe these are changing times. 

Author Response

Thank you very much for your kind review.

Reviewer 2 Report

The authors have made an interesting attempt at “Misinformation about the COVID-19 vaccine in the online Catholic media.” The manuscript is interesting; however, the authors need to justify the scientific writing manuscript. Some of the general comments are provided below:

1.     How is the high volume of misinformation regarding COVID-19 vaccines quantified or measured? Are there specific metrics or indicators used to assess the extent and impact of misinformation?

2.     What specific keywords were used in the algorithm to collect relevant publications related to the COVID-19 vaccine? How were these keywords determined, and were they adjusted or refined during the data collection process?

3.     What were the specific categories used for analysis in the spreadsheets and Google Forms? How were these categories determined, and were they based on existing frameworks or developed specifically for this research?

4.     Were 17 countries specifically selected based on certain criteria, or were they a result of the websites that were included in the study?

5.     In the qualitative analysis, were there any specific themes or patterns related to the COVID-19 vaccine that emerged across different countries or regions? Were there any notable differences in the content or narratives discussed in publications from different countries?

6.     Can you provide more details about the content and characteristics of the opinion publications? Did they primarily discuss the COVID-19 vaccine from a religious perspective or did they cover a broader range of topics?

7.     How were the findings regarding the formats of opinion publications interpreted or discussed in relation to the study's objectives? Did the prevalence of column formats or editorial formats have any implications for the dissemination of vaccine-related information within the Catholic community?

8.     Did the study investigate the influence or impact of the different publication formats on readers' attitudes or beliefs towards the COVID-19 vaccine? If not, were there any recommendations or suggestions for future research in this area?

9.     Did the study investigate whether the authors' profiles had any influence on the reception or credibility of their opinion publications within the Catholic community or among readers in general?

10. Did the study examine whether the authors' profiles had any correlation with the tone, stance, or persuasiveness of the opinion publications? Did certain profiles tend to have a stronger impact on readers' attitudes or beliefs?

11. Did the study assess the accuracy and reliability of the information presented within the thematic categories, particularly in relation to the topic of COVID-19 vaccines? Were there any instances of misinformation or misleading content identified?

12. Were there any limitations or challenges in categorizing the publications into thematic categories? How did the study address any ambiguities or complexities in the classification process?

Author Response

Thank you for your feedback. Please find the answers to your questions and answers below.

1. This research quantified the number of headlines that link to abortion and the percentage of news stories that cite religious sources when talking about the vaccine. In this research, we analyzed different aspects of the construction of the news event around the covid-19 vaccine in Catholic media, but we did not have the role of verifiers of the information, so we do not perform a generalizable quantification, but we provide indicators for the study of misinformation.

2. The keywords used with the search algorithms to retrieve the publications were:

coronavirus

coronavirus vaccine

Covid

covid 19 vaccin

covid-19 vaccin

pandemia

pandemia vaccino

pandemic

pandémie

vaccin

vaccin covid

vaccinazione covid

vaccine

vaccines

vaccini

vaccino

Vaccino anti Covid

vaccino covid

vaccins

vacina

vacina covid

vacina covid-19

vacinação covid-19

vacinas

vacuna

vacuna contra la covid-19

vacuna covid

vacuna covid 19

vacuna covid-19

vacuna de la COVID-19

vacunación

vacunación covid

vacunas

vacunas contra la COVID-19

3. The categories used for the analysis of the spreadsheets and forms are detailed in the following tables of the article: Table 2. Categories for gender analysis of publications; Table 2: Categories for headline analysis; and Table 4. Categories for publication analysis (as a whole). The source has been incorporated in the headline of each publication to clarify the origin of the publications.

4. This research was part of a project studying misinformation in the 5 chosen languages. The countries are the result of algorithmic searches for keywords in the 5 languages.

5. In this research, comparisons were made by language, but not by country or region. It would certainly be interesting to consider comparisons by region or country in future research.

6. At the time of designing the research, different options were considered, among which were either to conduct a content analysis of each publication, or to conduct a category analysis of the format, style and thematic content. In this particular research, it was decided to carry out a categorization to analyze variables related to the format, topic and sources of the publications, taking into account the objectives of the research project in which it is framed. Undoubtedly, it would be interesting to expand the research in the future to conduct in-depth content analysis and thus be able to answer questions such as the one posed.

7. This approach was not part of the present research, although we understand that it opens up a very interesting line of future research.

8. This study did not focus on the impact of different publication formats on readers' attitudes. It did not focus on news consumption but on news production. It would certainly be a good avenue for future research in this area.

9. The present study did not analyze the impact of each author on the recognition of his or her publications, a line of research that we consider interesting to explore in the future. We appreciate this comment which opens up a new avenue of research.

10. The study did not specifically examine whether author profiles had any correlation with the tone, stance or persuasiveness of opinion publications. Exploring the possible impact of author profiles on readers' attitudes or beliefs is an interesting avenue for future research as it would provide valuable insights into the role of authorship in shaping readers' perceptions and opinions.

11. The present research was part of a project where there was another organization in charge of fact-checking, data that we did not have available for this paper. 

12. Yes, difficulties were encountered during the classification process of some publications. When this occurred, the strategy of cross-checking was applied, which consists of involving more than one person in the analysis of the same publication. In this way, possible ambiguities or discrepancies due to subjective interpretations or different perspectives were resolved. We incorporate this explanation in the article, as well as citing the source necessary to justify the use of this strategy.

You can find the suggestions made in the new version of the manuscript. 

Warm regards, 

Reviewer 3 Report

The authors present an important, well described research analysis of publications about the COVID-19 vaccine in Catholic media outlets in 5 languages.  Their delineation and quantification of types of writing, such as informative vs. opinion, and relation to newsworthy events are important.

While I appreciate the detail of 22 figures, I think many of the results may be better shown in tables or combined, for example, Figures 3-4, 6-7, 8-9, 10-11, 12-13, 14-15, 16-17, 18-19, and 21-22 show related data by country (as I list them here with the '-', they seem to be the most directly related).   A table by language may be able to show much this data in a more succinct fashion, with figures reserved for cases where a table doesn't show the data well enough. 

In particular, Figures 6, 10, 12, 14, 16, and 21 are simply binary or tertiary outcomes, which are then broken down by language in each figure that follows.  This is the case of many of the figures.

In other cases, data for a timeline is very nicely represented by a figure (e.g., Figure 5, note that 'publications' is misspelled.)

The discussion, to my read beginning at line 481, tends to repeat much of the results without offering much interpretation.  I would suggest removing the repetitive citation of figures that were mostly addressed in the results.   Some of the material from the well-written introduction could be moved to the discussion in order to relate the findings of the current study to published literature.  

in Lines 135-138, several 'Error! Reference source not found' noted.

Typographical errors as noted in the review.

Author Response

Thank you very much for your comments and suggestions. Following your recommendation, we have reduced the number of figures and corrected the mistake in Figure 5. We removed the label "Number of publications" as we realized it was not accurately positioned on the axis and deemed it unnecessary. Regarding the error in the mentioned reference, we reviewed the document sent and we see them working correctly, we do not know if there is an error in the document downloaded or sent from Vaccines. Regarding the discussion, we added interpretation and reflection and linked the results with the theoretical discussion.

Reviewer 4 Report

This is a well written paper, on a "hot" subject, and it describes  an aspect that is seldom highlighted, namely publications that are faith-based. This journal is the right place for this article, even if it does not describe biological or medical aspects of vaccines. If the article is curtailed, the dates of publication are less important for conveying the message.

The only problem is the number of figures. I suggest that figures 1-5 should be included, and the others should be described in the "results" section of the text.

English language is adequate.

Author Response

Thank you very much for your useful feedback. We have reduced several figures and kept those considered necessary to present and understand the research in the best possible way and also take into consideration the suggestions of other reviewers.

Warm regards,

Round 2

Reviewer 2 Report

The authors have modified the articles and responded accordingly, it is now acceptable for publication.